# Study of 4,4'-Methylene Diisocyanate Phenyl Ester-Modified Cassava Residues/Polybutylene Succinate Biodegradable Composites: Preparation and Performance Research

**Lijie Huang \*, Hanyu Zhao, Hao Xu, Shuxiang An, Chunying Li, Chongxing Huang, Shuangfei Wang, Yang Liu and Jie Chen**

College of Light Industry and Food Engineering, Guangxi University, Nanning 530004, China
\* Correspondence: jiely165@gxu.edu.cn

**Abstract:** Biomass materials have become a research focus for humankind, due to the decreasing availability of fossil fuels and the increasing release of greenhouse gas. In this work, we prepared biodegradable composites with waste cassava residues and polybutylene succinate (PBS) by modifying cassava residues using 4,4'-methylene diisocyanate phenyl ester (MDI) and tested their properties. The effects of MDI modification on the structure, mechanical properties, water absorption, microstructure, and thermal stability of the composites were studied via Fourier transform infrared spectroscopy, contact angle measurement, mechanical property testing, water absorption analysis, scanning electron microscopy, and thermogravimetric analysis, respectively. The results showed that the tensile strength and flexural strength of the material increased by 72% and 20.89%, respectively, when the MDI-modified cassava residue content was 30%. When 10% MDI-modified cassava residues were added, the tensile strength increased by 19.46% from 16.96 MPa to 20.26 MPa, while the bending strength did not change significantly. The water contact angle of the MDI-treated cassava residues exceeded 100°, indicating excellent hydrophobicity. Thus, MDI modification can significantly improve the mechanical properties and thermal stability of the biocomposite. The composites were immersed in distilled water for 96 h. The water absorption of the cassava residues/PBS composite was 2.19%, while that of the MDI-modified cassava residues/PBS composite was 1.6%; hence, the water absorption of the MDI-modified cassava residues/PBS composite was reduced to 26.94%. This technology has wide application potential in packaging, construction, and allied fields.

**Keywords:** 4,4'-methylene diisocyanate phenyl ester; cassava residues; polybutylene succinate; biodegradable; modification

## 1. Introduction

Cassava, which is originally from the Amazon basin, is known as the "King of Starch" and is the sixth most important food crop in the world, providing a food source for the survival of nearly one-tenth of the world population [1,2]. Wei et al. studied the dry flesh of cassava and reported that its starch contents are 76.4–87.0%, protein contents are 2.5–3.3%, soluble sugar contents are 2.7–6.8% and cellulose contents are 2.2–3.7% [3]. A total of 250–300 t cassava tubers were processed for starch, which produced 1.6 t of cassava peels and 280 t of cassava residues with 85% water (15% dry matter only). China annually produces 1.5 million t of cassava residues, the main components of which are natural macromolecular compounds such as starch, cellulose, hemicellulose, lignin, a small amount of protein, and other trace elements. Moreover, cassava residues are biodegradable, reasonably priced, and non-toxic [4–8]. Figure 1 shows photographs of cassava, cassava starch, and cassava residues.

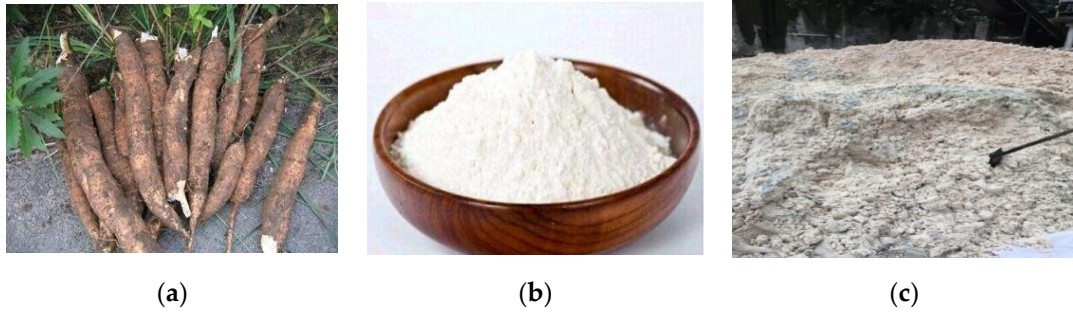

|        |        |        |
|:------:|:------:|:------:|
| (**a**) | (**b**) | (**c**) |

**Figure 1.** Photographs of (**a**) cassava, (**b**) cassava starch, and (**c**) cassava residues.

Meanwhile, polybutylene succinate (PBS) is a new, green, and eco-friendly material that has good mechanical properties, good heat and chemical resistance, and complete biodegradability. PBS can also be completely burned without producing any toxic gas [9–11].

Liminana et al. used almond shell powder and PBS to prepare composite materials and employed different compatibilizers to enhance the properties of these composites. Their results showed that all compatibilizers could enhance the ductile properties of the composite materials [12]. Cheng used ramie and PBS to prepare composite materials and investigated their mechanical performance. After plasma treatment, the tensile strength first increased, then decreased with increasing addition of ramie; the flexural strength showed the same tendency [13]. Huang et al. used sugarcane rind fiber and PBS to prepare composite materials and studied the biodegradability. The weight loss and mechanical property degradation were higher when 5 wt% sugarcane rind fiber was added after being buried in soil for 100 days [14]. Hongsriphan et al. used ammonium sulfate as a fertilizer and prepared composite films from lemon basil particles and PBS via hot pressing. The films were buried in agricultural land; ammonium sulfate was dissolved by adsorbed water and diffused into the surrounding soil as a fertilizer. Their study indicated considerable toughness reduction and rigidity improvement of the composite films compared to those of pure PBS films [15]. Previous research has shown that some fibers and PBS can be used to prepare composite materials; however, these raw materials are expensive, and their processing technology is complicated.

This work demonstrated a simple process to prepare the valuable composites of PBS and cassava residues. The composites have the advantages of cost-effectiveness and improved disintegrability of PBS. Moreover, the composites maintain good tensile strength (21–24 MPa) and flexural strength (28–32 MPa). The –NCO group in 4,4'-methylene diisocyanate phenyl ester (MDI) reacts with the hydroxyl (–OH) group on the surface of cassava residues to generate a carbamate bond, which reduces the surface polarity of the cassava residues. Thus, the hydrophilic cassava residues become compatible with the hydrophobic PBS and the interfacial adhesive force between the cassava residues and PBS is increased. These facilitate the transfer of interfacial stress between the fiber and PBS polymer matrix, thus imparting good mechanical properties to the composites. Herein, the composites were prepared by extruding and pressing a mixture of several raw materials. To some degree, this alleviates the environmental pollution caused by non-degradable plastics. In addition, it improves the performance of biodegradable composites, aids in their development, and helps lower the costs of the composites, considering that 1 t of cassava residue costs 900 yuan (approximately $129.71). The composite materials could be applied in packaging, construction (for example, wood flooring) and other fields [11], thus offering broad and significant application prospects.

## 2. Materials and Methods

### 2.1. Materials

Cassava residues were obtained from Wuming Anning Starch Co., Ltd., Guangxi Province, China. PBS (injection molding grade) was bought from Anqing Hexing Chemical Corporation Limited, Anhui

Province, China. Potassium bromide (analytical reagent) was purchased from Chengdu Jinshan Chemical Reagent Co., Ltd., Sichuan Province, China. MDI (chemically pure) was purchased from Shanghai Aladdin Bio-Chem Technology Co., Ltd., China. Acetone (analytical reagent) was purchased from Chengdu Kelon Chemical Reagent Factory, Sichuan Province, China.

### 2.2. Analysis of Chemical Composition of Cassava Residues

The holocellulose of the raw material was tested according to GB/T 2667.10-1995 [16]. In addition, the lignin was tested according to GB/T 2677.8-1994 [17] and GB/T 10337-2008 [18]. The starch was tested according to GB/T 5009.9-2016 [19], and the ash content was tested according to GB/T 742-2008 [20].

### 2.3. MDI-Modified Cassava Residues

MDI (1 g) was dissolved in 160 mL of acetone. The cassava residues were passed through a 60 mesh sieve and mixed with the MDI-acetone solution in a solid-to-liquid ratio of 1:8 in an Erlenmeyer flask under constant stirring; the mass ratio of MDI and the cassava residues was 1:20. The resulting mixture was placed in a constant-temperature water bath at 70 °C for 4 h and stirred every half hour to prevent the cassava residues from sticking to the Erlenmeyer flask. The mixture was then allowed to stand until cooling, washed 3–5 times in acetone, and suction-filtered using a water-circulating vacuum pump to remove the residual MDI. After the reaction, the cassava residues were placed in a fume cupboard and the acetone was evaporated. Thereafter, the mixture was dried to a constant weight in an electric heating drum wind drying oven at 70 °C and then cooled to room temperature. The obtained samples were stored in sealed bags.

### 2.4. Composite Preparation

Different amounts of MDI-modified cassava residues (5, 10, 15, 20, and 30 wt%) and PBS powder were combined in a high-speed mixer for 30 min. The mixed samples (80 g per sample) were placed into a mold (15 cm × 15 cm), hot-pressed at 130 °C under a pressure of 10 MPa for 10 min in a flat vulcanizer (XLB25-D, Qingdao Jinrunqi Rubber Machinery Co., Ltd., Jiao Nan Shi, China), and then cooled at room temperature for 15 min before demolding.

### 2.5. Performance Analysis of Composite Material

#### 2.5.1. FTIR Analysis

The samples were ground into a powder and passed through a 200-mesh sieve. Next, the powder was dried for 6 h in an electric drum wind drying oven. The samples were then mixed with KBr powder (mass ratio of 1:100) in an agate mortar and ground next to an infrared dryer until blended. Subsequently, a tableting machine pressed the samples for 3 min under pressures of 35 MPa to 37 MPa and compressed the samples into a transparent, smooth round thin sheet. The samples were analyzed using a Fourier transform infrared (FTIR) spectrometer (TENSOR II, Bruker, Billerica, MA, USA) with pure KBr as the test background at a resolution of approximately 4 cm$^{-1}$ in the wavenumber range of 4000 cm$^{-1}$ to 400 cm$^{-1}$.

#### 2.5.2. Contact Angle Analysis

The MDI-modified cassava residues were ground into a powder and passed through a 200-mesh sieve. Next, the powder was dried until the quality was consistent. Then, a tableting machine was used to compress the samples for 3 min under a pressure of 36 MPa, yielding a uniform, smooth wafer. Finally, 3 μL of ultrapure water was dropped on the sample surfaces using a microsyringe, and the samples were analyzed using a Kruss contact angle measuring instrument (DSA100, Kruss, Hamburg, Germany) by capturing images at regular time intervals and measuring the contact angles.

### 2.5.3. Determination of Mechanical Properties

The mechanical properties of the MDI-modified cassava residues/PBS biodegradable composites were measured using an electronic universal material testing machine (3367, Instron, Norwood, MA, USA). The tensile strength was tested according to the GB/T 1040.2-2006 [21] standard at a tensile rate of 10 mm/min. The length, width, and thickness of the samples were 15 cm, 2 cm, and 3 mm, respectively. Five samples were each tested once, and the measurements were averaged. The flexural strength test was performed according to the GB/T 1449-2005 [22] standard with a chuck reduction rate of 2 mm/min and a span of 30 mm. The length, width, and thickness of the samples were 15 cm, 2 cm, and 3 mm, respectively. Five samples were each tested once, and the five measurements were averaged.

### 2.5.4. Water Absorption Analysis

Prior to the water absorption test, the MDI-modified cassava residues/PBS biodegradable composites were dried to a constant weight at 65 °C and cooled to room temperature. Then, the samples were immersed in distilled water for 12 h. The water absorption c was calculated as follows:

$$c = (m_2 - m_1)/m_1 \times 100\% \tag{1}$$

where $m_2$ is the wet weight of the composites (g), and $m_1$ is the dry weight of the composites (g). Three samples from each group were tested, each sample was tested once, and the measurements were averaged.

### 2.5.5. Surface Morphology Analysis

A spline of the composite material was dried in an oven at 65 °C for 8 h. The fracture surface of this spline and the surface of a spline of the composite material after water absorption were subjected to spray-gold treatment. Then, the spline was fixed on a sample stage with a conductive adhesive, and the cross-sectional morphology and the binding state were observed by scanning electron microscopy (SEM, F16502, Phenom-World BV, Eindhoven, The Netherlands).

### 2.5.6. Biodegradability Analysis

The biodegradability was tested according to the GB/T 19275-2003 [23] standard by a burial test, and the rate of mass loss of the MDI-modified cassava residues/PBS biodegradable composites was studied. The length, width, and thickness of the samples were 15 cm, 2 cm, and 3 mm, respectively. The samples were dried at 65 °C until the weight was constant. The samples were then weighed and buried at a depth of 10 cm to 15 cm in an experimental plot (Xixiangtang District, Nanning City, Guangxi Province, China) for 10, 20, 30, 40, and 60 d. The total nitrogen content of the soil was 1.256 g/kg, and the available phosphorus content was 31.068 mg/kg. During the burial test, the temperature was 12–34 °C and humidity was 70–80%. After retrieving the samples, their surfaces were cleaned with absolute ethanol and a hairbrush. The samples were again dried at 65 °C until the weight was constant. For each group, three samples were each tested once, and the measurements were averaged. After washing and drying, 5 mg or 10 mg of the sample (sheet or grains of the samples) was placed in an alumina crucible, and its thermal stability was analyzed using a synchronous thermal analyzer (STA449F5, Netzsch, Selb, Germany) over a temperature range of 30–700 °C at a heating rate of 10 °C/min under a nitrogen flow rate of 20 mL/min. The mass loss rate was calculated as follows:

$$c = (m_1 - m_2)/m_1 \times 100\% \tag{2}$$

where $m_1$ and $m_2$ are the weight of the composites before and after burial (g), respectively.

### 2.6. Statistical Methods

All the data represented the results per independent experiment. The data were expressed as the mean ± standard deviation (SD). Microsoft Excel (Microsoft Corp., Redmond, WA, USA) was used for calculations of the SD. The figures were drawn with Origin 8.0 (OriginLab Corp., Northampton, MA, USA).

## 3. Results and Discussion

### 3.1. Chemical Composition of Cassava Residues

Table 1 shows the main chemical composition of the cassava residues. The holocellulose content was 45.76%, lignin content was 4.98%, starch content was 47.27% (used for alcoholic fermentation), and ash content was 1.72%. In addition, the cassava residues contained a little cassava peel, fat, and protein. The fat, protein and other components accounted for approximately 0.27% [24].

**Table 1.** Main chemical composition of cassava residues.

| Chemical Composition | Holocellulose | Starch | Lignin | Ash Content |
|---|---|---|---|---|
| Mass fraction/% | 45.76 | 47.27 | 4.98 | 1.72 |

### 3.2. FTIR Analysis of Cassava Residues before and after Modification

Figure 2a,b show the FTIR spectra of the unmodified and MDI-modified cassava residues, respectively.

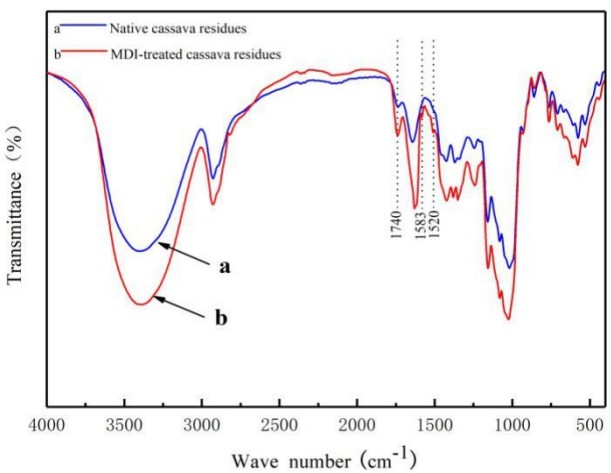

**Figure 2.** FTIR spectra of cassava residues.

The broad peak at 3402 cm$^{-1}$ originating from the stretching vibrations of the –OH group indicated that the cassava residues contained many polar –OH groups [6] and exhibited high water absorption. The stretching vibration absorption peak of methyl or methylene in cellulose or hemicellulose appeared at 2929 cm$^{-1}$, while the stretching vibration peak of hydroxyl in esters or organic acids appeared at 1737 cm$^{-1}$. The characteristic peak of absorbed water in the raw material appeared at 1645 cm$^{-1}$. Compared with curve a, the absorption peak positions changed in curve b. After MDI treatment of the cassava residues, a sharp absorption band appeared at 1740 cm$^{-1}$ arising from the hydroxyl in the carbamate bond [25,26]. New characteristic peaks appeared at 1583 cm$^{-1}$ and 1520 cm$^{-1}$ corresponding to the vibration of the benzene ring skeleton and bending vibration of N-H, respectively [25,27]. These characteristic peaks indicated the formation of a carbamate bond from the reaction of the –NCO group in MDI with the –OH groups on the surface of the cassava residues, which facilitated MDI grafting onto the cassava residues.

### 3.3. Contact Angle Analysis

The samples were analyzed using a contact angle measuring instrument by capturing the image of the water droplet every 5000 ms and measuring the water contact angle. The variation in contact angle with time is depicted in Figure 3. Figure 3 shows that the ultra-pure water droplets (0 ms) dropped onto the surface of both unmodified and MDI-modified cassava residue samples maintain a hemispherical shape. However, the water contact angles on the unmodified and MDI-treated cassava residues were 76.9° and 103.9°, respectively. In addition, the water contact angle of the unmodified cassava residues varied significantly with time. Between 5000 ms and 25,000 ms after dropping the water droplet onto the unmodified cassava residues, the water contact angle became 0°. This was because the cassava residues contained many surface hydroxyl groups and were strong absorbents.

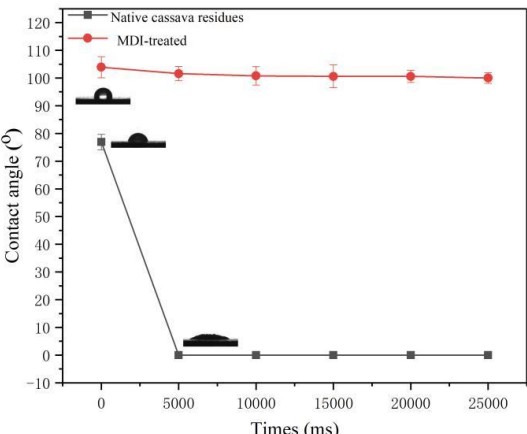

**Figure 3.** Variation in contact angle with time.

In contrast, the water contact angle on the MDI-treated cassava residues changed only slightly with time. The water contact angle of the MDI-treated cassava residues was over 100°, which indicated excellent hydrophobicity. The reaction of the –NCO group in MDI with the –OH groups on the cassava residue surface reduced the amount of surface –OH groups on the cassava residues, thus rendering them hydrophobic.

### 3.4. Mechanical Property Analysis of Cassava Residues after Modification

The tensile strength and flexural strength of the MDI-modified cassava residues/PBS biodegradable composites were measured using an electronic universal material testing machine, and the results are shown in Figure 4.

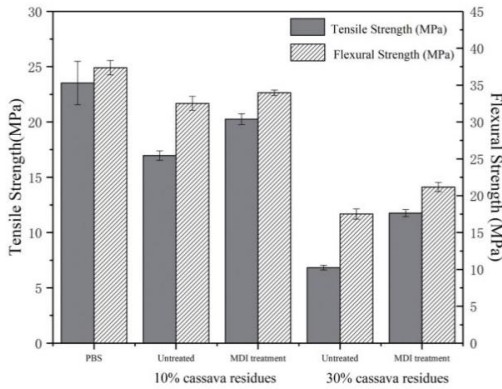

**Figure 4.** Effect of 4,4′-methylene diisocyanate phenyl ester (MDI) modification of cassava residues on mechanical properties of composites.

The tensile and flexural strengths of the MDI-treated cassava residues/PBS composites increased by various degrees, in contrast with those of the unmodified cassava residues/PBS composites, as shown in Figure 4. With the addition of 10% MDI-treated cassava residues, the tensile strength increased by 19.46% from 16.96 MPa to 20.26 MPa, while the bending strength did not change significantly. When 30% MDI-treated cassava residues were added, the tensile strength increased by 72%, and the bending strength increased by 20.89%. These increases can be attributed to the –NCO group in MDI reacting with the –OH groups on the surface of cassava residues, which produced a carbamate bond that reduced the surface polarity of the cassava residues. Thus, the hydrophilic cassava residues became compatible with the hydrophobic PBS, increasing the interfacial adhesive force between the cassava residues and PBS. This facilitated the transfer of interfacial stress between the fiber and the polymer matrix, and imparted good mechanical properties to the composites.

### 3.5. Water Absorption Analysis of Cassava Residues after Modification

The composite with 10% MDI-treated cassava residue content was immersed in distilled water for 12–96 h, and the water absorption of MDI-treated cassava residues/PBS composite was studied.

The results are shown in Figure 5. Evidently, the water absorption gradually increased and reached a steady-state with prolonged immersion time for both the unmodified and MDI-treated cassava residues/PBS composites. When the composites were immersed in distilled water for 96 h, the water absorption of the cassava residues/PBS composite was 2.19% and that of the MDI-modified cassava residues/PBS composite was 1.6%. Thus, the water absorption of the MDI-modified cassava residues/PBS composite was reduced by 26.94%. The water absorption of the composites and the hydrophilicity of the cassava residues were directly related, whereas the hydrophilicity, surface chemical composition of the cassava residues, and the interfacial chemical properties of composites were all closely related. In the MDI-modified cassava residues, the isocyanate group in MDI replaced the –OH in the surface of the cassava residues, which decreased the amount of surface –OH groups in the cassava residues. Moreover, the hydrophobic constituents were transferred to the cassava residues, thus greatly increasing their hydrophobicity. Chemical modification, which reduced the surface polarity of the cassava residues, improved the interfacial compatibility between the cassava residues and PBS. Thus, the water between cassava residues and PBS reduced, which decreased the water absorption of the composites.

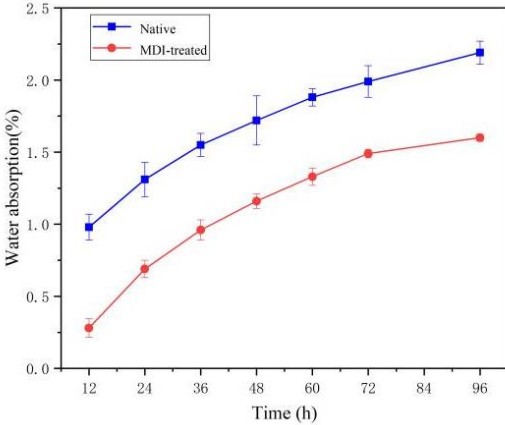

**Figure 5.** Effect of MDI modification on water absorption of 10% cassava residues/polybutylene succinate (PBS) biodegradable composite.

### 3.6. SEM Analysis of Cassava Residues before and after Modification

Figure 6 shows the SEM images of the tensile fracture surfaces of the composites with 10% cassava residues before and after MDI modification.

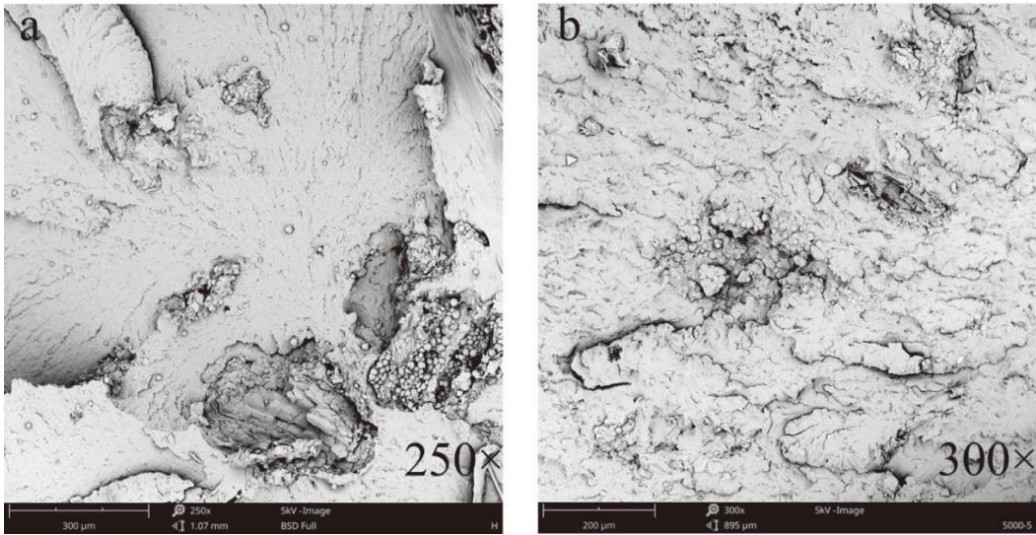

**Figure 6.** Tensile fracture surfaces of composite (**a**) before MDI modification and (**b**) after MDI modification.

As shown in Figure 6, in contrast with the composite containing unmodified cassava residues, the tensile fracture surface of the MDI-treated cassava residues/PBS composite was uniform and smooth, with no noticeable gap between the cassava residues and PBS. As can be seen in Figure 6, under external forces, the unmodified cassava residues of the composites were rooted out, whereas the MDI-treated cassava residues of the composites were fractured near the roots. The MDI-treated cassava residues and PBS had good interfacial compatibility, implying that the PBS matrix sufficiently infiltrated the grooves on the surface of the MDI-treated cassava residues. Moreover, the interfacial adhesive force between the modified cassava residues and PBS was high [28]. Thus, the MDI modification of cassava residues improved the interfacial compatibility between the cassava residues and PBS, and the strong interaction force in the two-phase area of contact improved the mechanical properties of the composite.

*3.7. Disintegrability Analysis of Cassava Residues before and after Modification*

The MDI-treated cassava residues were used to improve the binding force between the cassava residues and PBS. The disintegrability of cassava residues/PBS composites with and without MDI treatment were studied, and the results are shown in Figures 7–9.

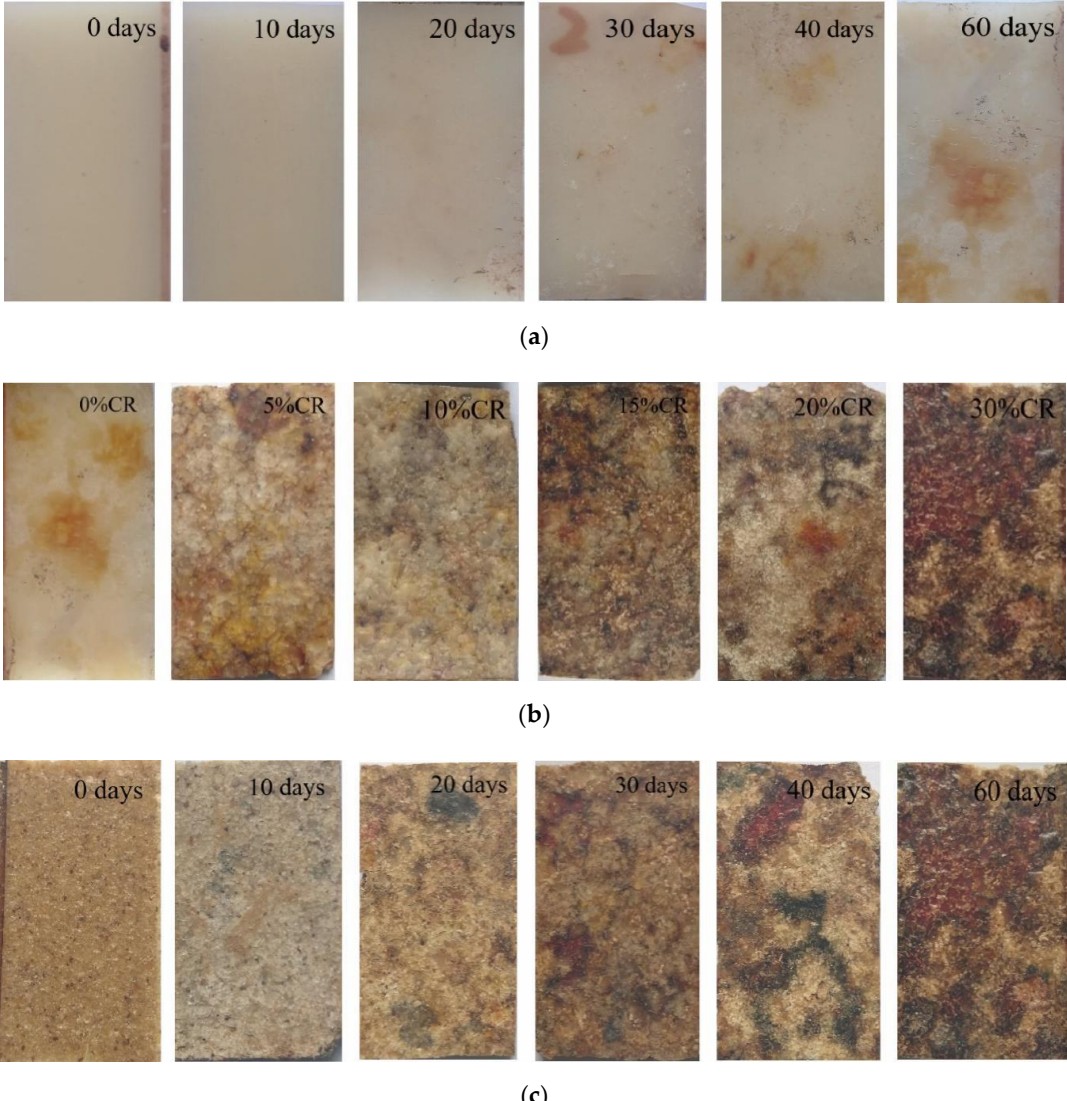

**Figure 7.** Surface morphologies after disintegration. (**a**) Pure PBS; (**b**) different cassava residue contents; (**c**) 30% cassava residue/PBS.

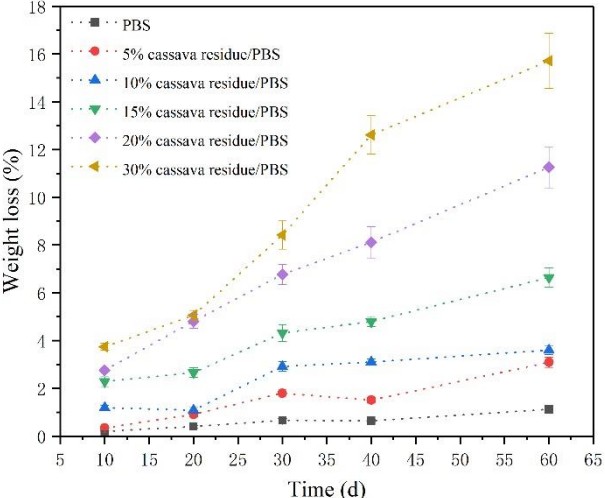

**Figure 8.** Weight loss of pure PBS and cassava residue/PBS composite.

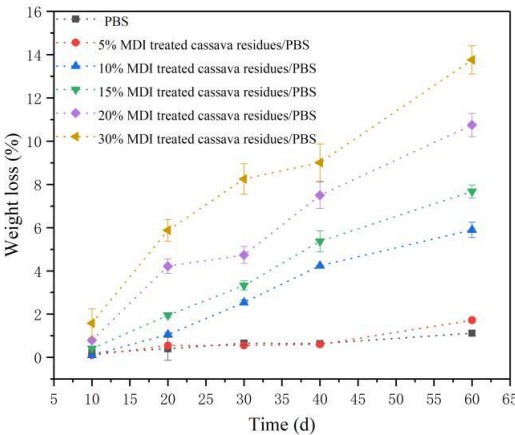

**Figure 9.** Weight loss of pure PBS and MDI-modified cassava residues/PBS composite.

As shown in Figure 7a, the sample surfaces developed dark brown plaques over time, showed little peeling, and became increasingly rough. As shown in Figure 7b, the sample surfaces of the composites changed significantly. With increasing content of the cassava residues, the color of the sample surfaces gradually deepened, and there were many large dark brown spots on the surface. Particularly, when the content of cassava residues was 30%, the degree of spots was the deepest. As shown in Figure 7c, the disintegrability of the samples improved with extended burial time. Based on the changes in the surface morphologies and weight loss, the disintegrability of the 30% cassava residues/PBS composite was improved compared to that of pure PBS. As shown in Figure 8, the disintegrability of the samples improved with extended burial time. After 60 d of disintegration, pure PBS showed a mass loss rate of 1.12%, the composites containing 10% cassava residues exhibited mass loss rates not more than 4%, and the composites containing 30% cassava residues exhibited a mass loss rate of 15.72%, which was 14 times that of pure PBS (1.12%). However, the samples disintegrated more easily than pure PBS when cassava residues were added. Thus, the 30% cassava residues/PBS composite showed better disintegrability compared with pure PBS.

Figure 9 shows that after 60 d of disintegration, the composites containing different amounts of MDI-treated cassava residues had mass loss rates of 1.72%, 5.9%, 7.68%, 10.75%, and 13.76%. Therefore, the samples disintegrated more easily when more MDI-treated cassava residues were added. In addition, the compatibility between the cassava residues and PBS was improved, which reduced the infiltration of water and microorganisms into the materials [29], resulting in a low mass loss rate of the samples. Over time, material corrosion and disintegration helped increase the mass loss rate of the 30% MDI-modified cassava residues/PBS composite to 13.76%, which is 12 times that of pure PBS (1.12%). A combined analysis of Figures 8 and 9 showed that the curves of mass loss rates before and after modification were very similar, which indicated that the MDI modification did not considerably affect the disintegrability of the composites.

Furthermore, as seen in Figure 10, after 60 d of disintegration, the surface integrity of the materials was severely damaged. As shown in Figure 10a, the sample surfaces of PBS changed significantly. After 60 d of disintegration, the erosion to the PBS surface of the lipase secreted by the microorganisms continued to increase [29], so that the number of pores on the surface of the composites continued to increase and spread to the entire surface of the composites. As shown in Figure 10b, after 60 d of disintegration, the internal cassava residues were exposed; at the same time, the grooves on the surface of the composites became wide. These phenomena were caused by the erosion of the composites by microorganisms in nature and the swelling and expansion of cassava residues in the composites [29]. As shown in Figure 10c, the sample showed a rough surface at the beginning of disintegration due to its high content of cassava residues, and the pores in the composites made it easy for microorganisms to enter the interior. After 60 d of disintegration, the internal cassava residues were exposed; at the same time, the grooves on the surface of the composites became wider. Many large grooves were

observed. Therefore, in a natural environment, it can be inferred that cassava residues and the surface PBS were first decomposed by microorganisms, following which the composites were layered and degraded according from the outside to the inside [30].

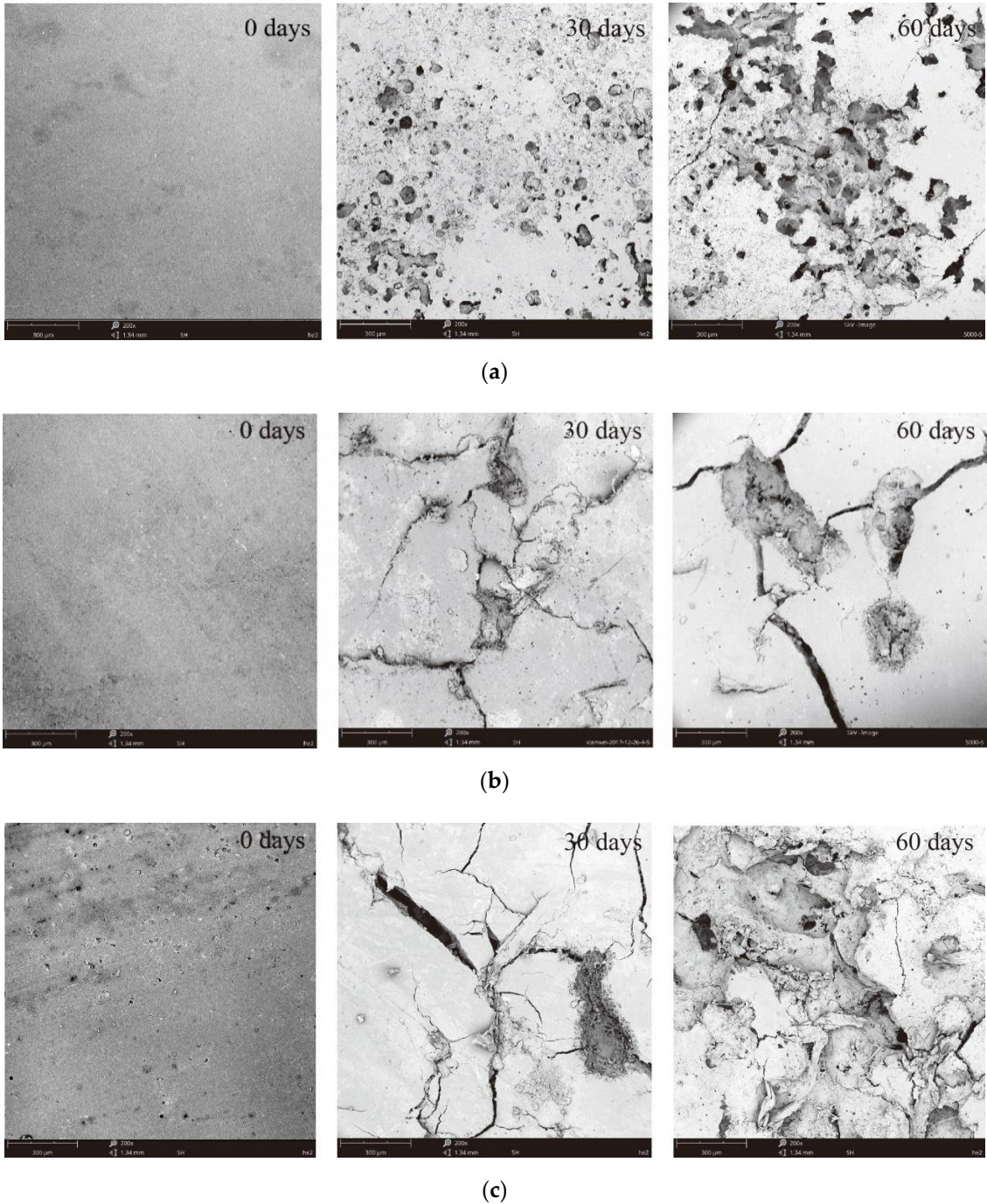

**Figure 10.** SEM images of (**a**) pure PBS, (**b**) 10% cassava residues/PBS composite and (**c**) 30% cassava residues/PBS composite after different disintegration times. (**a**) Pure PBS; (**b**) 10% cassava residues/PBS composite; (**c**) 30% cassava residues/PBS composite.

After disintegration, the thermal stability of the composites was tested using a synchronous thermal analyzer, and the results are shown in Figure 11. As can be seen from the derivative thermogravimetry (DTG) curve in Figure 11, after disintegration, the thermal decomposition of the composites in the synchronous thermal analyzer occurred via two steps: The thermal decomposition

of cassava residues in composites (near 325 °C), and the thermal decomposition of PBS (near 403 °C), which were consistent with the two maximum weight loss rate peaks shown in DTG. However, after 60 d of disintegration, the grading phenomenon of the thermal disintegration of composites weakened, and the first weight loss rate peak began to flatten out during DTG, which indicated that in a natural environment, the microorganisms first decomposed the cassava residues, followed by PBS disintegration.

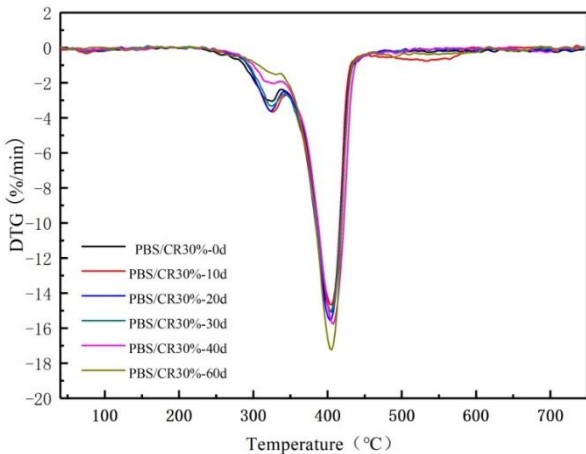

**Figure 11.** Derivative thermogravimetry (DTG) curves after disintegration of composites.

## 4. Conclusions

The water contact angle of MDI-treated cassava residues exceeded 100°, which illustrated their excellent hydrophobicity. After modifying the cassava residues with MDI, the interfacial compatibility between the cassava residues and PBS improved, and the MDI-treated cassava residues/PBS composite appeared uniform and smooth. The two materials exhibited good surface adhesion as well as a strong interfacial adhesion force at the two-phase area of contact, which improved the mechanical properties of the composites. When 10% MDI-treated cassava residues were added, the tensile strength increased by 19.46% from 16.96 MPa to 20.26 MPa, while the bending strength did not change significantly. When 30% MDI-treated cassava residues were added, the tensile strength increased by 72%, and the bending strength increased by 20.89%. The samples disintegrated more easily when more MDI-treated cassava residues were added. After 60 d of disintegration, the 30% cassava residues/PBS composite exhibited a mass loss rate of 15.72%, which was 14 times that of pure PBS (1.12%). Over time, the initial disintegration temperature of the samples gradually increased disintegration. The microorganisms first decomposed the cassava residues, following which PBS was disintegrated. In this work, the composite fully utilized the cassava residues, which is a waste product, thus demonstrating a new application area for cassava residues to mitigate environmental or air pollution. However, the mechanisms underlying the modification of cassava residues, the improved mechanical properties, and the disintegration of cassava residues and PBS have not yet been clarified, and are expected to be areas of future research.

**Author Contributions:** Data curation, L.H.; formal analysis, L.H. and H.Z.; investigation, S.A.; methodology, S.W.; project administration, L.H.; resources, H.X. and J.C.; software, H.Z. and H.X.; supervision, L.H., C.L. and Y.L.; validation, L.H., H.Z. and C.H.; writing—original draft, H.Z.; writing—review and editing, L.H.

**Funding:** This research was funded by the Guangxi Key Laboratory of Clean Pulp and Papermaking and Pollution Control (No. ZR201806-6); Construction Project of Characteristic Specialty and Teaching Base (Center) of Experimental Training of Undergraduate Universities in Guangxi from 2018 to 2020(T3050094101).

**Acknowledgments:** The authors are grateful for the financial support from the Guangxi Key Laboratory of Clean Pulp and Papermaking and Pollution Control (No. ZR201806-6); Construction Project of Characteristic Specialty and Teaching Base (Center) of Experimental Training of Undergraduate Universities in Guangxi from 2018 to 2020(T3050094101).

**Conflicts of Interest:** The authors declare no conflict of interest. The sponsors had no role in the design, execution, interpretation, or writing of the study.

**Abbreviations**

MDI: 4,4'-methylene diisocyanate phenyl ester; PBS: polybutylenes succinate.

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
