# Peer review of "Study of 4,4‘-Methylene Diisocyanate Phenyl Ester-Modified Cassava Residues/Polybutylene Succinate Biodegradable Composites: Preparation and Performance Research"

_processes, doi:10.3390/pr7090588_

Round 1

Reviewer 1 Report

The manuscript is clearly structured in Introduction, Analysis, Conclusion and Bibliography. The text is comprehensible and concentrates on the essential results. The abstract presents the essential statements of the article in a logical context.

This paper is worth of publication in the Journal of Processes, if the following changes/revisions can be applied/addressed:

- statistical methods need to be explained and error bars and stats model should be added.

- more lit review needs to be conducted regarding the similar biological materials

Reviewer 2 Report

In this work, the authors prepared composites of cassava residue (0-30% wt/wt) and polybutylene succinate (PBS) (70-100% wt/wt). To increase the hydrophobicity, the cassava residue was pretreated with 4, 4'-methylene diisocyanate phenyl ester (MDI). Only one dosage of MDI was used and little information on the reaction extent was revealed. The composites of cassava residue and PBS were examined in multiple tests, showing the changes in material properties and improvement in soil degradation. The information may be interesting on biodegradable plastics and composites.  

There is a concern on the method.  A large amount of acetone was used in the process. Would the solvent affect the economical feasibility and greenness of the technology?  In addition, acetone was mixed with cassava residue under the reaction conditions. Did acetone change the properties of cassava residue?  Some controlled samples such as acetone-treated cassava residue (without MDI) should be tested and included.    

The introduction should be revised to justify why cassava residue was used to make PBS composite. What unique properties could it contribute? If the goal was aimed at reducing the cost and/or improving the biodegradability of PBs, can the content of cassava residue be increased, say, to more than 30% wt/wt? What was the limiting factor?   

Special notes:

L40: “Every 250 tons to 300 tons cassava residue processed, there are solid surface of 1.6 tons, and 280 tons cassava residues of 85% water content”. This sentence is confusing.  Does it mean 250 to 300 tons of cassava (dry matter) being processed? What is the solid surface? How could the surface be measured in weight?  Does it mean generation of 280 tons of cassava residues with 85% water (25% dry matter only)? 

L61: “used ammonium sulfate as a fertilizer, and they prepared composite films from lemon basil particles and PBS via hot pressing.” It is confusing. Why was fertilizer ammonium sulfate used for preparation of composite films? 

L63: “higher toughness reduce and rigidity improve of the composite films”. Was the toughness of the composite films reduced or increased? Was this property change compared to pure PBS or something else?

L66: “there is little research on preparing composite materials from cassava residue and PBS”. It is not a good justification because little work was done.  Does it reduce the cost or improve special material properties of the composites?  

L68: “This work proposes a simple process to use cassava residue of to produce highly valuable.” The sentence should be revised. This work demonstrated a simple process to make valuable composites of PBS and cassava residue.  

L99:”solid-to-liquid ratio of 1:8”. Was it the mass ratio? Please give exact mass ratio of MDI and cassava residue. Under best conditions, how many OH groups could be reacted? 

L156:” The samples were dried in the Xixiangtang District, Nanning City, Guangxi Province, China”. What does it mean?

L159: “After retrieving the samples…” Could the samples be degraded into small fragments? Were all small pieces of a sample recovered?  

Figure 4:  The material strength of pure PBS should be included. Please provide error bars of the measurements.  

Figure 5: Please provide error bars of the measurements.  

Figure 6: Please indicate the composites of (a) and (b).

Figure 7: Where is the figure?

L272: “the contrast of pure PBS and 30% cassava residue/PBS composite is the 30% cassava residue/PBS composite biodegradability of the samples improved”. The sentence is somehow wordy and confusing. Was the biodegradability improvement concluded based on the weight loss or surface change?

Figure 8: Please indicate the samples of (a) and (b). The Y scale of weight loss is not visible.  

L287: “the water run-through or microorganisms between cassava residue and PBS reduced”. What is the water run-through? How could the microbes be seen in Figure 8?

L290:” Comparing the unmodified and MDI-modified cassava residue/PBS composites, the weight loss is almost the same.” Is this shown in Figure 8 or Figure 9?

Author Response

Dear Reviewer,

We thank you for the time and effort you spent reviewing our paper. We are pleased to note that you found our research work is interesting and environmentally very relevant We are extremely grateful for your pointing out several problems to help us improve the quality of our work.

In response to your comments, we have carefully reconsidered the layout of our work and have attempted to resolve a ll the problems you mentioned. In particular, the revised manuscript that we are resubmitting has been significantly improved in the following aspects.

Thank you for your consideration. I look forward to hearing from you.

Sincerely,
Lijie Huang
College of
Light Industry and Food Engineering
Guangxi University
Nanning 530004
China
[86 15077108861]
[86 0771 3276156]
[email protected]

Response to Reviewer 2 Comments:

In this work, the authors prepared composites of cassava residue (0-30% wt/wt) and polybutylene succinate (PBS) (70-100% wt/wt). To increase the hydrophobicity, the cassava residue was pretreated with 4, 4'-methylene diisocyanate phenyl ester (MDI). Only one dosage of MDI was used and little information on the reaction extent was revealed. The composites of cassava residue and PBS were examined in multiple tests, showing the changes in material properties and improvement in soil degradation. The information may be interesting on biodegradable plastics and composites.

1.There is a concern on the method. A large amount of acetone was used in the process. Would the solvent affect the economical feasibility and greenness of the technology? In addition, acetone was mixed with cassava residue under the reaction conditions. Did acetone change the properties of cassava residue? Some controlled samples such as acetone-treated cassava residue (without MDI) should be tested and included.

Response: The solvent did not affect the economical feasibility and ecological viability of the technology. Acetone has the advantage of low cost and can be recycled and reused after modification.modification. Cassava residue did not dissolve in acetone; therefore, the properties of cassava residue did not change due to acetone treatment. Acetone is only a solvent for MDI. In Figure 2, which shows the FTIR spectra of TIR spectra of native native cassava residuescassava residues and MDIand MDI--treated cassava residuestreated cassava residues, we can see that no new absorption peak appears; thus, the chemical properties of cassava residue do not change by acetone treatment. Moreover, XRD analysis proved that acetone has almost no effect on the crystallization of cassava residue cellulose; please see the supplementary information Figure S1Figure S1 for details. Some samples such as native cassava residue (without MDI) have be tested and included, please see the supplementary information Figure SFigure S33--S9S9 for details.

Supplementary information see the attachment.

2. The introduction should be revised to justify why cassava residue was used to make PBS composite. What unique properties could it contribute? If the goal was aimed at reducing the cost and/or improving the biodegradability of PBs, can the content of cassava residue be increased, say, to more than 30% wt/wt? What was the limiting factor?

Response: The introduction has been revised to explain why cassava residue was used to make the PBS composite. (Please see L73 in the revised manuscript for details). The unique properties of the composite included reduced cost and improved the composite included reduced cost and improved disintegrability of PBS. Moreover, the composites maintained good tensile strength (21over, the composites maintained good tensile strength (21--24 MPa) and flexural strength 24 MPa) and flexural strength (28(28--32 MPa). 32 MPa). (Please see L74-75 in the revised manuscript for details). When the content of cassava residue reached 40%, the tensile strength of the composite greatly reduced which significantly reduced the application value.

Special notes: L40: “Every 250 tons to 300 tons cassava residue processed, there are solid surface of 1.6 tons, and 280 tons cassava residues of 85% water content”. This sentence is confusing. Does it mean 250 to 300 tons of cassava (dry matter) being processed? What is the solid surface? How could the surface be measured in weight? Does it mean generation of 280 tons of cassava residues with 85% water (25% dry matter only)?

Response: The The sentence has been changed to “A total of 250has been changed to “A total of 250––300 t of cassava tubers were 300 t of cassava tubers were processedprocessed for starchfor starch, which produces 1.6 t of cassava peels and 280 t of cassava residues , which produces 1.6 t of cassava peels and 280 t of cassava residues with 85% water with 85% water (1(15% dry matter only5% dry matter only))”. ”. (revised manuscript L40-42).

L61: “used ammonium sulfate as a fertilizer, and they prepared composite films from lemon basil particles and PBS via hot pressing.” It is confusing. Why was fertilizer ammonium sulfate used for preparation of composite films?

Response: The author of this paper reported thatThe author of this paper reported that “These fillers acted as nucleating agents in order to increase the degree of crystallinity and the rigidity of the composite films. The films were buried on agricultural land; ammonium sulfate was dissolved by adsorbed water and diffused into surrounding soil as a fertilizer.” (revised manuscript L64-69).

(Hongsriphan, N; Pinpueng, A. Properties of Agricultural Films Prepared from Biodegradable Hongsriphan, N; Pinpueng, A. Properties of Agricultural Films Prepared from Biodegradable Poly (Butylene Succinate) Adding Natural Sorbent and Fertilizer. J. Polym. Environ, 2019, 27, Poly (Butylene Succinate) Adding Natural Sorbent and Fertilizer. J. Polym. Environ, 2019, 27, 11--10.10.)

L63: “higher toughness reduce and rigidity improve of the composite films”. Was the toughness of the composite films reduced or increased? Was this property change compared to pure PBS or something else?

Response: The The sentence has been changed to “Their study indicatehas been changed to “Their study indicated considerable toughness d considerable toughness reducreductiontion and rigidity improvement of the composite films and rigidity improvement of the composite films compared to compared to those of those of pure PBS pure PBS filmfilmss””. . (revised manuscript L67-69).

L66: “there is little research on preparing composite materials from cassava residue and PBS”. It is not a good justification because little work was done. Does it reduce the cost or improve special material properties of the composites?

Response: The composite reduces cost and improves material properties. The sentence “there is little research on preparing composite materials from cassava residue and PBS” has been deleted. Also, the following text has been added: “The composite has the advantages of cost effectiveness and improved disintegrability of PBS. Moreover, the composites maintained good tensile strength (21-24 MPa) and flexural strength (28-32 MPa).” revised manuscript L73-75).

L68: “This work proposes a simple process to use cassava residue of to produce highly valuable.” The sentence should be revised. This work demonstrated a simple process to make valuable composites of PBS and cassava residue.

Response: The sentence has been changed to “ This work demonstrated a simple process to prepare valuable composites of PBS and cassava residue s ..” (revised manuscript L71-72).

L99:”solid-to-liquid ratio of 1:8”. Was it the mass ratio? Please give exact mass ratio of MDI and cassava residue. Under best conditions, how many OH groups could be reacted?

Response: A total of 1 g cassava residues was added to 8 mL solution of MDI-acetone, and the exact mass ratio of MDI and cassava residue is 1:20.” (Please see L109-112 in the revised manuscript for details). Under optimal conditions, the reaction rate of OH groups is 10%. MDI mainly reduces the water absorption of the cassava residues by reacting with the hydroxyl groups on the surface of the cassava residue or adhering to the surface of the fiber and improves the compatibility between the cassava residues and the PBS matrix. With the addition of 5% MDI, the surface of the cassava residues is evenly coated, and the performance will be considerably improved. However, with increasing MDI addition, the excess MDI will no longer react with the fiber, but attached to the cassava residues in the form of solid deposits. This adhesion is not strong and will seriously affect the fiber bonding properties. In addition, excess MDI may agglomerate in the composite material, destroying the uniformity inside the composite material. When subjected to external forces, the stress cannot be distributed uniformly and rapidly in the material, resulting in stress concentration on the stressed part, collapse of the composite system, deterioration of mechanical properties, and easy fracture. This observation was similar to the research results by Yu Xueling et al.

(Yu, X. L.; Song, J. B.; Yang, W. B.; Yang, J. F. Effects of KH-550 content on properties of tea stems flour/HDPE. J. For. Environ. 2015, (1),92-96. )

L156:” The samples were dried in the Xixiangtang District, Nanning City, Guangxi Province, China”. What does it mean?

Response: The word “dried” was incorrect in this context. The word has been changed to buried”. (revised manuscript L170).

L159: “After retrieving the samples…” Could the samples be degraded into small fragments? Were all small pieces of a sample recovered?

Response: The samples did not disintegrate into small fragments in 60 days of disintegrable experiments. During the 60-day experiment, the material did not break into fragments, but the surface changed. However, due to moisture absorption and disintegration, the strength of the material was very poor. If the disintegration time was prolonged, it could have been disintegrated into fragments, which limits complete sample collection and calculation of an accurate mass loss rate.

Figure 4: The material strength of pure PBS should be included. Please provide error bars of the measurements.

Response: The material strength of pure PBS has been included in Figure 4. The error bars have been provided in the Figure. Please see the revised manuscript for details.

Figure 5: Please provide error bars of the measurements.

Response: The error bars have been included in Figure 5. Please see the revised manuscript for details.

Figure 6: Please indicate the composites of (a) and (b).

Response: Composites of (a) and (b) have been indicated in Figure 6.

Figure 7: Where is the figure?

Response: Figure 7 has been included.

L272: “the contrast of pure PBS and 30% cassava residue/PBS composite is the 30% cassava residue/PBS composite biodegradability of the samples improved”. The sentence is somehow wordy and confusing. Was the biodegradability improvement concluded based on the weight loss or surface change.

Response: The sentence has been changed to “Based on the changes in surface morphologies and weight loss, the disintegrability of 30% cassava residues/PBS composite was improved as compared to that pure PBS.” (revised manuscript L298-299).

Figure 8: Please indicate the samples of (a) and (b). The Y scale of weight loss is not visible”.

Response: I have indicated the samples of (a) and (b). The Y axis of weight loss has been defined.

L287: “the water run-through or microorganisms between cassava residue and PBS reduced”. What is the water run-through? How could the microbes be seen in Figure 8?

Response: We apologize for the incorrect word choice. The relevant literature reports that “The composites films, however, show alterations in their surfaces after 30 days of test. Black spots, probably associated with the formation of biofilms that occurs due to t he branching of hyphae and mycelia, are observed on the surface of these films. Thus, we can see that “the compatibility between cassava residue and PBS was improved, which reduced the infiltration of water and microorganisms into the materials. materials.” (revised manuscript L323 324).

(Thainá Araújo de Oliveira, Islaine de Oliveira Mota, Francisco Edinaldo Pinto Mousinho, Renata Barbosa, Laura Hecker de Carvalho, Tatianny Soares Alves. Biodegradation of mulch films from poly(butylene adipate co terephthalate), carn auba wax, and sugarcane residue. J. Appl. Polym. Sci. 2019, 48240.)

L290:” Comparing the unmodified and MDI-modified cassava residue/PBS composites, the weight loss is almost the same.” Is this shown in Figure 8 or Figure 9?

Response: The sentence has been revised to “Combined analysis of Figure 8 and Figure 9 showed that the curves of mass loss rates before and after modification were very similar, which indicated that the MDI modification did not considerably affect the disintegrability of composites.” (revised manuscript L328 330).

Reviewer 3 Report

In this manuscript, Huang and coauthors reported the use of MDI to modify cassava residues (CR); the MID-modified CR was further studied in the context of CR/PBS composite. I would recommend the authors consider my comments below:

1. None of the data shown in the manuscript have error bars, including Fig. 3, Fig. 4, Fig. 5, Fig. 8 and Fig. 9. Without error bars, the claimed increase in properties, such as “72% increase in tensile strength” and “20.89% increase in flexural strength”, would become meaningless.

2. There is no caption for Fig. 7.

3. The x-axis of Fig. 8 is missing. Without it, there is no way to make a comparison between Fig. 8 and Fig. 9.

4. In Line 300, the statement “These peaks indicate that in a natural environment, microorganisms first decompose the cassava residue, and then, PBS is degraded.” Is severely wrong. How could results obtained from thermal degradation tell us any information about biodegradation?

5. There are no scale bars in Fig. 6 and Fig. 10.

6. The authors are recommended to perform a thorough grammatical check throughout the text. Many sentences in the manuscript do not make sense, to name a few:

Line 40: Every 250 tons to 300 tons cassava residue processed, there are solid surface of 1.6 tons, and 280 tons cassava residues of 85% water content.

Line 44: Moreover, cassava residue is which are biodegradable, price reasonable, and non-toxic [4–8].

Line 68: This work proposes a simple process to use cassava residue of to produce highly valuable.

Line 173: According to these contents, the rich chemical composition of cassava residue has bright prospects.

Author Response

Dear Reviewer,

We thank you and the reviewers for your thoughtful suggestions and insights. The manuscript has benefited from these insightful suggestions. I look forward to working with you to move this manuscript closer to publication in the Processes.

The manuscript has been rechecked and the necessary changes have been made in accordance with the reviewers’ suggestions. The responses to all comments have been prepared and given below.

Thank you for your consideration. I look forward to hearing from you.

Sincerely,
Lijie Huang
College of Light Industry and Food Engineering
Guangxi University
Nanning 530004
China
[86 15077108861]
[86 0771 3276156]
[email protected]

Response to Reviewer 3 Comments

In this manuscript, Huang and coauthors reported the use of MDI to modify cassava residues (CR); the MID-modified CR was further studied in the context of CR/PBS composite. I would recommend the authors consider my comments below:

1. None of the data shown in the manuscript have error bars, including Fig. 3, Fig. 4, Fig. 5, Fig. 8 and Fig. 9. Without error bars, the claimed increase in properties, such as “72% increase in tensile strength” and “20.89% increase in flexural strength”, would become meaningless.

Response: The error bars have been included in Figs. 3, 4, 5, 8, and 9. Please see the revised manuscript for details.

2. There is no caption for Fig. 7.

Response: The caption for Fig. 7 has been included. Please see the revised manuscript for details.

3. The x-axis of Fig. 8 is missing. Without it, there is no way to make a comparison between Fig. 8 and Fig. 9.

Response: The x axis of Fig. 8 has been added. Please see the revised manuscript for details.

4. In Line 300, the statement “These peaks indicate that in a natural environment, microorganisms first decompose the cassava residue, and then, PBS is degraded.” Is severely wrong. How could results obtained from thermal degradation tell us any information about biodegradation?

Response: The previous statement is not exact, which has been modified to “ But after 60 d of disintegration, the grading phenomenon of the thermal disintegration of composites weakened, and the first weightlessness rate peak began to flatten out in DTG, w hich indicate d that in a natural environment, microorganisms first decompose d the cassava resid ue s , followed by PBS disintegration. ” Please see the revised manuscript L375-378 for details.
The DTG curves of cassava residue and PBS were provided in the supplementary information Figure S2 for details.

Supplementary information see the attachment.

5. There are no scale bars in Fig. 6 and Fig. 10.

Response: The scale bars have been included in Figs. 6 and 10. Please see the revised manuscript for details.

6. The authors are recommended to perform a thorough grammatical check throughout the text. Many sentences in the manuscript do not make sense, to name a few: Line 40: Every 250 tons to 300 tons cassava residue processed, there are solid surface of 1.6 tons, and 280 tons cassava residues of 85% water content.

Response: The sentence has been changed to “A total of 250–300 t of cassava tubers were processed for starch, which produced 1.6 t of cassava peels and 280 t of cassava residues with 85% water (15% dry matter only).” (revised manuscript L40-42).

Line 44: Moreover, cassava residue is which are biodegradable, price reasonable, and non-toxic [4–8].

R: The sentence has been changed to “This work proposes a simple process to prepare valuable composites of PBS and cassava residues.” (revised manuscript L71-72).

Line 173: According to these contents, the rich chemical composition of cassava residue has bright prospects.

Response: This sentence has been deleted. (see the revised manuscript L192-193).

Round 2

Reviewer 2 Report

The revised manuscript addressed the concerns of this reviewer. I have no further questions. 

Reviewer 3 Report

The authors have addressed my comments. Though with reluctance, I am in favor of its publication on Processes. However, a grammatical check is recommended. 
